# Identification of novel non-myelin biomarkers in multiple sclerosis using an improved phage-display approach

Andrea Cortini[1☯], Sara Bembich[1☯], Lorena Marson[1], Eleonora Cocco[2], Paolo Edomi [1]*

**1** Department of Life Sciences, University of Trieste, Trieste, Italy, **2** Multiple Sclerosis Center, University of Cagliari/ATS Sardegna, Cagliari, Italy

☯ These authors contributed equally to this work.
* edomi@units.it

**Data Availability Statement:** All relevant data are within the paper and its Supporting Information files.

## Abstract

Although the etiology of multiple sclerosis is not yet understood, it is accepted that its pathogenesis involves both autoimmune and neurodegenerative processes, in which the role of autoreactive T-cells has been elucidated. Instead, the contribution of humoral response is still unclear, even if the presence of intrathecal antibodies and B-cells follicle-like structures in meninges of patients has been demonstrated. Several myelin and non-myelin antigens have been identified, but none has been validated as humoral biomarker. In particular autoantibodies against myelin proteins have been found also in healthy individuals, whereas non-myelin antigens have been implicated in neurodegenerative phase of the disease.

To provide further putative autoantigens of multiple sclerosis, we investigated the antigen specificity of immunoglobulins present both in sera and in cerebrospinal fluid of patients using phage display technology in a new improved format. A human brain cDNA phage display library was constructed and enriched for open-read-frame fragments. This library was selected against pooled and purified immunoglobulins from cerebrospinal fluid and sera of multiple sclerosis patients. The antigen library was also screened against an antibody scFv library obtained from RNA of B cells purified from the cerebrospinal fluid of two relapsing remitting patients. From all biopanning a complex of 14 antigens were identified; in particular, one of these antigens, corresponding to DDX24 protein, was present in all selections. The ability of more frequently isolated antigens to discriminate between sera from patients with multiple sclerosis or other neurological diseases was investigated. The more promising novel candidate autoantigens were DDX24 and TCERG1. Both are implicated in RNA modification and regulation which can be altered in neurodegenerative processes. Therefore, we propose that they could be a marker of a particular disease activity state.

## Introduction

Multiple sclerosis (MS) is a chronic neuroinflammatory disease of the central nervous system (CNS) characterized by the presence of areas of demyelination and neuroaxonal loss (lesion)

**Funding:** The research was funded by the Italian multiple sclerosis foundation (Fondazione italiana sclerosi multipla – FISM, www.aism.it) grant n. 2002R26 and 2004R6 to PE and by internal support of the University of Trieste. The funder had no role in study design, data collection and analysis, decision to publish, or preparation of the manuscript. There was no additional external funding received for this study.

**Competing interests:** Andrea Cortini, Sara Bembich and Paolo Edomi are listed as inventors on PCT patent applications titled Biomarkers for the diagnosis of multiple sclerosis (PCT/EP20 11/ 072440). Other authors declare that they do not have competing interests related to this study. There are no additional patents, products in development or marketed products to declare. This does not alter our adherence to all the PLOS ONE policies on sharing data and materials, as detailed online in the guide for authors.

in the brain and spinal cord [1]. Although the etiology of the disease is still not clear MS is widely accepted to be an autoimmune disease with both the innate and adaptive immune system playing a role in promoting inflammation, demyelination and neurodegeneration which lead in turn to the disruption of neuronal signalling [2].

Whereas the infiltration of autoreactive T cells mounting an aberrant response against CNS autoantigens is a well-established step in MS pathogenesis, the role of B cells in this process is still to be fully elucidated. Nevertheless, B cells are considered to be involved in pathological mechanisms particularly for their role as antigen presenting cells and for the production of pro-inflammatory cytokines [3,4]. In addition, autoantibodies from B cells could be implicated in MS pathogenesis or neurodegeneration. The presence of intrathecal antibodies produced by plasma-cells in 95% of the patients and the correlation of plasma cell percentage with disease progression strongly support the involvement of the antibody-mediated B cell mechanism in the pathogenesis of MS [5,6]. Sequence analysis of V segments of Igs from single-cell demonstrates that clonal expansion is a prominent feature in MS [7–9], suggesting that a specific antigen or group of antigens is driving CNS B-cell activity in MS. These evidences are further supported by the presence of lymphoid follicle-like structures, observed in the cerebral meninges of some MS patients [10,11]. In this context, numerous studies have tried to establish the targets of the humoral response in MS patients. However, up to date the antigen specificity of intrathecally synthesized oligoclonal IgG in MS is still unknown [2,12]. Since one of the most prominent features of MS is the presence of demyelination areas, in the past 40 years the attention has been concentrated mostly on myelin antigens such as myelin basic protein (MBP), myelin oligodendrocyte glycoprotein (MOG) and proteolipid protein (PLP). However, the data collected so far on the anti-myelin specific antibody response are contradicting and now, with several studies showing no difference between MS patients and healthy controls (HC), its contribution in the pathogenesis of MS seems unlikely [12]. Our understanding of MS has changed in the past years and is now well accepted that in MS the intersection of autoimmune and neurodegenerative processes contribute to disease pathogenesis [13–16]. Although the link between these two processes is still unclear, recent evidences have associated the neurodegenerative phase of the disease with the presence of antibodies against non-myelin antigens such as neurofilaments [17,18], neurofascin [19], RNA binding proteins [20] and potassium channels [21].

Due to the heterogeneity of MS and the phenomenon of epitope spreading a single associated antigen is unlikely. Nevertheless, the discovery of the antigens driving the B cell response in MS will be a crucial step to better understand its pathogenesis and to develop new diagnostic tests that are currently limited to magnetic resonance imaging (MRI) and the detection of oligoclonal Abs in the cerebrospinal fluid (CSF) of patients [22,23].

In this study, we report an improved experimental approach for the identification of new autoantigens in MS using antibody and cDNA phage display libraries. In particular, we provide evidence of the advantage of the use of single-chain variable fragment (scFv) library for the selection of new candidate autoantigens in MS. Furthermore, we identified and serologically validate two novel MS candidates showing their potential use as biomarkers for MS.

## Materials and methods

### Serum and CSF samples

All the serum and cerebrospinal fluid samples were kindly supplied by the Multiple Sclerosis Centre of the University of Trieste and Cagliari. All investigation were conducted according to the principles expressed in the Declaration of Helsinki and written informed consent was obtained. All the data of patients were anonymously conserved. The samples were stored at

**Table 1. Details of the cohorts used in the study.**

| Groups of samples | Sera/ CSF | n | Diagnosis | Female/Male | Mean Age in years |
|---|---|---|---|---|---|
| **Samples used for library selection** | | | | | |
| 1 | Sera | 90 | RR-MS | 71/19 | 40 |
| 2 | CSF | 11 | CIS (4) | 1/3 | 41 |
| | | | RR-MS (7) | 5/2 | 40 |
| **Samples used for ELISA validation tests** | | | | | |
| Cohort 1 | CSF | 12 | CIS (4) | 1/3 | 41 |
| | | | RR-MS (8) | 6/2 | 34 |
| | CSF | 8 | OND | 0/8 | 69 |
| Cohort 2 | CSF | 30 | RR-MS | 19/11 | 37 |
| | CSF | 33 | OND | 12/21 | 62 |
| Cohort 3 | Sera | 18 | RR-MS | 10/8 | 39 |
| | Sera | 19 | OND | 8/11 | 65 |

Data of the groups of patients samples: for each group the source (sera or CSF), the number (n), the diagnosis (RR-MS, CIS or OND), the sex ratio and the mean age of the samples are indicated.

-80°C. All the patients with multiple sclerosis had been diagnosed according to the McDonald criteria and the populations data are summarized in Table 1. The patients with other neurological diseases (OND) comprised: polyneuropathies, polyneuritides, polyradiculoneuritides, encephalitides, myelitides, meningitides, leukoencephalopathies, vasculitides, Miller-Fisher and Guillain-Barré syndromes, amyotrophic lateral sclerosis, spastic tetraparesis, paraneoplastic neuropathies, Charcot-Marie-Tooth syndrome, spinal cord injuries, hydrocephalus, subarachnoid haemorrhages. For the selection of the human brain library sera from 90 untreated relapsing–remitting (RR) MS patients (EDSS 0–6.5) were distinctly pooled and purified by protein G chromatography (HiTrap protein G HP, *GE Healthcare*) using FPLC (ÄKTA™ system). Samples from 7 untreated RR MS patients (EDSS 0–3.5) and 4 clinically isolated syndrome (CIS) patients (EDSS 0–2) were pooled for the selection against CSF samples. All samples used in ELISA for validation tests were from untreated patients.

## Bacterial strains

*Escherichia coli* bacterial strains used were: DH5α F' (Gibco BRL): F'/*endA1 hsd17* ($r_K^-m_K^+$) *supE44 thi-1 recA1 gyrA* (Nal$^r$) *relA1Δ* (*lacZYA-argF*) *U169 deoR* (F80*dlacD*-(*lacZ*)*M15*), for phage propagation and scFv library construction; TOP10F': F'{*lac*Iq*Tn10*(TetR)} *mcr*A Δ(*mrr-hsd*RMS-*mcr*BC) Φ80*lacZ*ΔM15 Δ*lac*X74 *rec*A1 *ara*D139 Δ(*ara-leu*)7697 *gal*U *gal*K *rps*L *end*A1 *nup*G, for cDNA human brain library construction; BS1365: BS591 F' Kan [BS591: *recA1 endA1 gyrA96 thi-1 D lacU169 supE44 hsdR17* (*lambda1mm434 nin5 X1-cre*)], for cDNA human brain library recombination; XL1-Blue: F'∷ Tn*10 proA⁺B⁺ lac1�q Δ (lacZ) M15/ recA1 endA1 gyrA96* (Nal$^r$) *thi hsdR17* ($r_K^- m_K^+$) *glnV44 relA1 lac*, for the expression of recombinant proteins; BL-21: F- ompT hsdSB (rB-mB-) gal dcm araB∷T7RNAP-tetA; HB2151: K12, *ara Δ(lac-pro), thi/F' proA⁺B⁺, lacI�q ΔZM15*, to produce soluble single-chain variable fragment (scFv).

## Construction of human brain phage-display peptide library

Human brain poly(A)+ RNA was purchased from Clontech (cod.6516-1) and used for the generation of the Open-Reading-Frame (ORF) selected cDNA phage display library using pEP2 vector as described previously [24]. Briefly, 1 μg poly(A)+ RNA and 2,5 μg of Random Primers

*Hin*dlll (Novagen) were mixed and first strand cDNA synthesis performed using 200 U of SuperScriptlll RT (lnvitrogen) and 200 U of RNaseOUT (lnvitrogen), in the presence of methylated dNTPs 0.5 mM (to protect internal restriction sites of *Eco*RI and *Hin*dlll from subsequent digestions), according to the manufacturer's instructions. The second strand was synthesized with 23 U of DNA polymerase I (Promega) and 0.8 U of RNase H (USB) incubating the reaction at 14˚C for 2 hours. After purification with phenol chloroform, cDNA was blunt ended by adding 1.5 U of T4 DNA polymerase (NEB), in the presence of 0.4 mM dNTP at 12˚C for 20 min, and ligated to the linker LINKPE (`5'- AGGGGAGGGGGCTTGAATTCA AGC-3'`) overnight at 16˚C. The cDNA fragments in the range 300–800 bp were gel-purified and amplified using primers ORIAMPEFOR (`5' GAGGGGGCTTGAATTCAAGC-3'`) and ORIAMPEREV (`5'- GGGGGCTTGAATTCAAGCTT-3'`) performing 30 cycles of amplification using 1 unit of Phusion Hot Start DNA Polymerase (Finnzymes). After purification, the amplified cDNA fragments were sequentially digested with *Hin*dIII (Promega) and *Eco*RI (NEB) and ligated to pEP2 vector using T4 DNA Ligase (NEB). The ligation mixture was purified and divided in 8 aliquots each of which was used to transform 40 µL of electrocompetent TOP10F′ *E. coli* cells (Invitrogen). The transformation mixture was plated on 25 µg/ml chloramphenicol and 12 µg/ml ampicillin to achieve ORF-selection; a dilution of transformed cells was plated only on chloramphenicol to estimate the library dimension. The resultant colonies were analysed by PCR, enzymatic fingerprinting (*Bst*OI) and sequencing to evaluate the library diversity. After selection on ampicillin plates, the *β-lactamase* gene was removed by trasforming BS1365 F' *E. coli* cells, which constitutively express *Cre* recombinase, with the phagemids of the ORF selected library, as previously described [24]. The phagemids produced by these bacteria were re-transformed in DH5αF' *E. coli*. The clones obtained represent the library of selected ORF fragments.

## Construction of single chain phage-display library

The scFv library was generated as described before [25] starting from total RNA obtained from a pool of B cells purified from the CSF of two relapsing remitting (RR) MS patients both positive for the presence of oligoclonal bands and without treatments: a female, age 40 (disease onset at 36), EDSS 3 and a male, age 29 (disease onset 27), EDSS 3.5. Briefly, total RNA was extracted from $2 \times 10^4$ B cells using PicoPure^TM RNA Isolation Kit (ARCTURUS) and First strand cDNA was synthesized using random hexamers and SuperScript^TM III RT (Invitrogen) following the manufacturer's protocol. Each family of VH and VL genes were amplified separately from the first strand cDNA by PCR using 1 unit of recombinant ExTaq Polymerase (TaKaRa) and specific VH primers or VL primers [26]. For the VH chains, the 3' primer was specific for the IgG subclass. Single VH and VL genes were combined to obtain two equimolar VH and VL mixtures, which were amplified and assembled as described by Sblattero et al [26]. The assembled PCR products coding for the scFv were then ligated into the phagemid vector pDAN5 and transformed into DH5αF' *E. coli* cells by electroporation. Positive cells were selected on ampicillin agarose plate. Single clones obtained from the transformation were analysed by PCR, enzymatic fingerprinting (*Bst*OI) and sequencing to evaluate the library diversity.

## Expression and purification of scFv antibodies

To produce recombinant scFv antibodies in soluble form, HB2151 cells were infected with phages of the antibody library from CSF and were grown at 37˚C in 2xYT medium containing 100µg/mL of ampicillin up to an OD equal to 0.5. After adding IPTG to a final concentration of 0.5 mM, growth was continued overnight. After centrifugation at 4500g for 20 min, the

pellet was resuspended in 10 ml of lysis buffer (Tris-HCl 20 mM pH 8.0, NaCl 500 mM, imidazole 5 mM, Triton X100 0.1%) per gram of cells, together with lysozyme 100 μg/mL and DNase 30 μg/ml, and incubated in ice for 60 min. Then the samples were centrifuged at 4500g for 20 min to separate the included bodies from the soluble cellular proteins. The included bodies containing the scFvs were resuspended in 10 ml of solubilizing buffer (Tris-HCl 20 mM pH 8.0, NaCl 500 mM, imidazole 5 mM, TritonX100 0.1% and urea 8M) and incubated for 1 hour at 4°C. The sample was centrifuged at 4500g for 20 min and the scFvs were purified by affinity chromatography using NiNTA resin (IBA). Folding of the scFvs was carried out directly on the column using a linear gradient of urea from 8.0 M to 0 M. The renatured scFvs were eluted using a buffer containing Tris-HCl 20 mM pH 8.0, NaCl 500 mM and imidazole 300 mM.

## Human brain phage library selection against pooled sera and CSFs

To obtain the phage particles for the panning process, the human brain library was grown in the presence of 25 μg/mL chloramphenicol and 1% glucose. After infection with M13 K07 helper phage, bacterial cells were grown in 25 μg/mL chloramphenicol and 25 μg/mL kanamycin overnight and the phages were PEG-purified as described previously. Briefly, for each panning, immunotubes (Nunc, Roskilde, Denmark) were coated with goat anti-human IgG Fc specific (Sigma, I2136) at a concentration of 10 μg/ml in phosphate-buffered saline (PBS: 0.01 M phosphate buffer, 0.0027 M KCl and 0.137 M NaCl, pH 7.4) at 4°C over-night. After washing immunotubes once with PBS, the tubes were blocked for 1h at room temperature (RT) with 2% BSA in PBS. Then, 0.5 ml of pooled sera (1:50 diluted in 2% BSA-PBS) or CSFs (1:3 diluted in 2% BSA-PBS) were added to the coated immunotubes and incubated for 90 minutes at RT. After washing the tubes 3 times with PBST 0.1% (PBS, 0.1% Tween 20 (v/v)) and 3 times with PBS, PEG–purified phages (approximately $10^{10}$), prepared from recombinated cDNA library, were diluted in an equal volume of 4% BSA-PBS, incubated for 30 minutes at RT and after added in the immunotubes for 30 minutes on a rotating platform, followed by 90 minutes of standing at RT. After 10 washes with PBST 0.1% and 10 with PBS, phages were eluted with 200 mM glycine pH 2.2–2% BSA and neutralized with 1M Tris-HCl pH 9. DH5α F' *E. coli* cells at O.D.$_{600nm}$ = 0.5 were infected with output phages for 40 minutes at 37°C. A dilution was plated on 2xTY agar plates containing chloramphenicol 25 μg/ml and glucose 1% for output titration; all the rest was infected with M13K07 helper phage (at a MOI of 20:1) and grown overnight in 10 ml of 2xYT with 25 μg/ml chloramphenicol and 25 μg/ml kanamycin, at 30°C. The cycle of selection was repeated two more times increasing the washes to 15 with PBST 0.1% and 15 with PBS for the second round and to 20 with PBST 0.5% and 20 with PBS for the last round. To monitor enrichment of specific clones, input and output phages from each round of selection were titrated and the ratio of output/input was determined. After selection, individual clones were respectively picked randomly and analyzed by PCR and subsequent enzymatic fingerprinting using *Bst*OI (Promega) and were screened for their reactivity by phage- ELISA. All shown human brain phage library clones were sequenced using the primer pEPSEQ.

## Phage-ELISA with serum and CSF

After selection, individual selected clones were grown in 96-well round-bottomed plates (Sarstedt) until to O.D.$_{600nm}$ = 0.5. Each clone was infected with M13K07 helper phage at a MOI of 20:1 at 37°C for 30 minutes and was left at 30°C overnight to allow the production of phages. 96-well flat-bottomed plates (Costar) were coated overnight at 4°C with 100 μl of goat anti-human IgG Fc specific, 10 μg/ml in PBS and blocked with 120 μl of 2% BSA-PBS for 1 h at RT.

Then, wells were incubated with pooled sera (diluted 1:50 in BSA 2%-PBS) or CSF (diluted 1:3 in BSA 2%-PBS) for 1 h at RT and subsequently washed 3 times with PBST 0.1% and 3 times with PBS. Fifty microliters of each supernatant containing the phages of the individual selected clones, grown overnight in the 96-well round-bottomed plate, were recovered, diluted in an equal volume of 4% BSA-PBS and incubated for 90 minutes at RT. M13K07 helper phage was used as internal control signal. After 3 washes with PBST 0.1% and 3 with PBS, 100 μl of a peroxidase conjugated anti-M13 monoclonal (Amersham/Pharmacia/Biotech), diluted 1:3000 in 2% BSA-PBS was incubated for 1 h at RT. After washing the plates 3 times with PBST 0.1% and 3 times with PBS, 65 μl/well of 3,3′,5,5′-tetramethylbenzidine (TMB, Sigma) was added and the color development stopped with 35 μl/well 1M $H_2SO_4$. The plates were read at 450 nm. For the successive phage-ELISAs performed on positive clones, the immunoreactivity for each phage clone (O.D. sample) was measured in relation to the internal control signal (M13K07 helper phage). A ratio of O.D. sample/O.D. helper > 5 was considered positive. Relatively to secondary phage-ELISAs, for each serum or CSF was calculated the ratio O.D. sample/O.D. helper.

## Human brain phage library selection against scFv phage library

For each cycle, a well of a microtitre plate (Costar) was coated with 9x10$^{12}$ phages of the CSF scFv library in carbonate buffer (NaHCO$_3$ 1M pH 9) at 4˚C overnight. After washing with PBS, saturation was carried out for 1 hour at room temperature with BSA 3% in PBS. The phages of the HB phage display library, purified by means of PEG as described before, were diluted in an equal volume of BSA 6% in PBS, incubated for 30 minutes at room temperature, added to the well and incubated for 30 min with stirring and a further 90 min, still at room temperature. Washing and elution were performed as for the selection with serum for a total of 3 cycle of selection. To monitor enrichment of specific clones, input and output phages from each round of selection were titrated and the ratio of output/input was determined. After selection, individual clones were respectively picked randomly and analyzed by PCR and subsequent enzymatic fingerprinting using *Bst*OI (Promega) and were screened for their reactivity by phage- ELISA. All shown human brain phage library clones were sequenced using the primer pEPSEQ.

## Phage-ELISA with soluble scFv

96-well flat-bottomed plates (Costar) were coated overnight at 4˚C with 100 μl of scFv CSF soluble form (purified as previously described), 5 μg/ml in PBS and blocked with 120 μl of 2% BSA-PBS for 1 h at RT. Phage ELISA on individual selected clones was then performed as described above.

## Production of the recombinant TCERG1 and DDX24

The cDNAs coding for the portions of the human protein Transcription Elongation Regulator 1 (TCERG1) (aa 677–1098) and DEAD-Box Helicase 24 (DDX24) (aa 100–924), were amplified by PCR from human brain total cDNA using specific primer designed on the basis of the sequences available in the NCBI database: (TCERG For: 5'-AAAATTCAGCTTTGATTTC AACGTGGGAGAAG-3' and TCERG Back 5'-TTCCTGCAGCCTTTTGTTGATGTGCT CCGTGG-3'; DDX24 for: 5'-TCATTGAATTCAATGAAGTTGAAGGACACAA-3' and DDX24 rev: 5'-ACCACCTGCAGGGTCCAGGTGGTGCTTCGGTG-3'). The cDNAs were gel-purified, digested with *Eco*RI and *Pst*I (NEB), ligated to the vector pASK-45plus (IBA) and transformed into Rosetta2 competent cells. Positive clones were analyzed by PCR using commercial available pASKfor and pASKrev primers. Positive clones for TCERG1 and for DDX24

were grown in 1 liter of 2xTYmedium containing ampicillin 50 μg/ml. At O.D.$_{600nm}$ = 0.5 production of the recombinant protein was induced by adding anhydrotetracycline 200 ng/ml and let the culture grow 3 hours at 37˚C. Recombinant TCERG1 was purified in two steps; after a first purification using the N-terminal Strep-tagII following the manufacture's protocol, a second purification was performed using the C-terminal His-tag. The Strep-tag purified sample was diluted 1:10 in solution A (20 mM Tris-HCl pH 8, 50 mM NaCl, 5 mM imidazole) and loaded on a NiNTA column (Amersham) equilibrated with the same buffer. After washing with 15 ml of solution B (20 mM Tris-HCl pH 8, 50 mM NaCl, 0.1% Triton X-100 (v/v), 20 mM imidazole) and 10 ml of solution A, the sample was eluted with 10 ml of elution buffer (20 mM Tris-HCl pH 8, 50 mM NaCl, 300 mM imidazole). To purify the recombinant DDX24 the bacterial pellet was resuspended with lysis buffer (10 ml/grams pellet) containing urea 6 M. The sample was then centrifuged at 5000 RPM for 20 minutes and the supernatants recovered and purified on "Ni-NTA" resin (*Amersham*). The resin equilibration was performed with 10 ml of solution A and urea 6 M; then the 0.2 μm- filtrated samples were passed twice on the columns. The washing steps were performed with progressive refolding of DDX24 using a linear gradient of urea from 6 M to 0 M. Elution was obtained with 10 ml of elution buffer.

## Serum and CSF ELISA

96-well flat-bottomed Reacti-Bind™ *Pierce* plates were coated overnight at RT with 1 μg of recombinant protein or 2 μg of synthetic peptide in PBS per well, an equivalent number of wells were coated with PBS only (background). The day after, wells were blocked with 200 μl of 2% BSA-PBS for 1 h at RT. Then, they were incubated for 1 h at RT with MS and OND sera, diluted 1:30; or MS and OND CSFs, undiluted, in final 2% BSA-PBS,. After 3 washes with PBST 0.05% and 3 with PBS, 100 μl of a peroxidase conjugated anti human IgG (*Dako*), diluted 1:300 in 2% BSA-PBS was incubated for 1 h at RT. After washing the plates three times with PBST 0.05% and three times with PBS, 65 μl/well of 3,3′,5,5′-tetramethylbenzidine (TMB, *Sigma*) was added and the color development stopped with 35 μl/well 1M H$_2$SO$_4$. The plates were read at 450 nm. For each O D. sample was subtracted the respective O.D. background.

## Statistical analysis

Statistical analysis was performed using *GraphPad Prism version 5.0*. The significance of between-group differences was assessed using unpaired *t* test (one-tailed), in case of both normal distributions, or using Mann-Whitney test (one-tailed), in case of non-normal distribution of at least one of the groups. A *p* value < 0.05 was considered statistically significant. ROC analysis was done to determine cut off values, sensitivity, specificity and LR+ of the tests.

# Results

## Production of an ORF-selected phage display cDNA library from human brain

We have previously described a novel system whereby, using oriented cloning of cDNA fragments upstream the gene for the *β-lactamase* followed by selection with ampicillin, we were able to generate ORF-enriched cDNA phage display libraries using the phagemid vector pEP2 [24]. In this work, we have used human brain (HB) total mRNA to generate a cDNA library of 6x10$^5$ independent clones. After selection on ampicillin a collection of 1.35x10$^5$ putative in-frame independent clones were obtained. A first step of quality control to assess the library diversity was performed on 20 randomly picked clones by PCR and enzymatic fingerprinting with *Bst*OI revelling an observed diversity of 100%. To further investigate the quality of the HB

library, 15 randomly picked clones were sequenced and identified through data bank screening, on http://blast.ncbi.nlm.nih.gov/ (S1 Table). Both cytoplasmic and nuclear protein were represented in the library which included, as expected, proteins specifically expressed by the nervous tissue such as "γEnolase", the "Neurofilament 3", the "Brain expressed X-linked 1" and the "Vesicle-associated membrane protein 1". On the basis of the sequencing data, the "in frame" clones represent the 87% (13/15) of the total (S1 Table).

## Identification of background clones

The HB library generated was used to identify new putative autoantigens and biomarkers for multiple sclerosis by selection of the phage library with Ig of MS patients. To optimally expose the antigen binding site, IgG of MS patients were immobilized on a solid support using an anti-human IgG-Fc specific monoclonal antibody. Therefore, to identify background phage clones, eventually recognized by the monoclonal antibody, a preliminary screening using only the anti-human IgG antibody was performed. After three rounds of selections on the HB library, 40 clones were screened and only three antigens specifically enriched by the anti-human antibody were observed (S2 Table). All these three antigenic specificities were considered, when resulting in the successive selections, as background clones deriving from the interaction with the monoclonal antibody used as coating and excluded from the results.

## Selections of the HB library with IgG from CSF of MS patients

A hallmark of MS is the presence of oligoclonal Abs in the CSF of patients due to the clonal expansion of Ab-secreting B cells in the CNS. To identify the epitopes recognized by these antibodies we performed three different biopanning of the HB phage library as summarized in the schematic overview of Fig 1. Initially we selected the HB phage library with a IgG pool of 11 CSFs collected from MS patients with a primary diagnosis of the disease (7 RR and 4 CIS) as described in Methods. Three rounds of selection were performed and at the end of each round the titer of the output phages was measured. The progressive amplification of interactive clones was evaluated by calculating the ratio between the output of the third round over the output of the first round (the input phage number was fixed for each round): an enrichment index of 10-fold was observed. To identify the higher responsive clones among the enriched ones, 50 CSFs clones from the last round of selection were randomly picked and individually assessed by phage-ELISA using the M13K07 helper phage as negative control. Out of 50 clones analysed 24 (48%) had an O.D. value 5 times higher than the helper phage O.D. and were therefore considered positive. All positive clones were then further characterized by PCR and enzymatic fingerprinting, sequenced, and the translated protein sequence determined. After sequence homology analysis, the 24 positive clones have been divided into 6 distinct antigenic groups (excluding background clones), each of them is represented by one or more overlapping clones. The summary of the antigens identified and the frequency of related clones is shown in Table 2 (column CSF). The blastN and blastP analysis performed for each antigen against the human reference RNA sequences (refseq_rna) and the human reference protein database (refseq_protein) are shown in S3 and S4 Tables respectively. Apart from background or out-of-frame clones, the clones matching a nucleotide, but not an aminoacid, sequence, were classified as "unidentified antigens", even if they correspond to an "in-frame" sequences without stop codons. If the nucleotide and aminoacid sequences identified by Blast analysis of the same clone correspond to different genes, the clone can be considered a mimotope. In all other cases, corresponding to 33% (8 clones out of 24), the clones carried a real ORF sequence. Among these, the most interesting antigen was DEAD-Box Helicase 24 (DDX24) which was

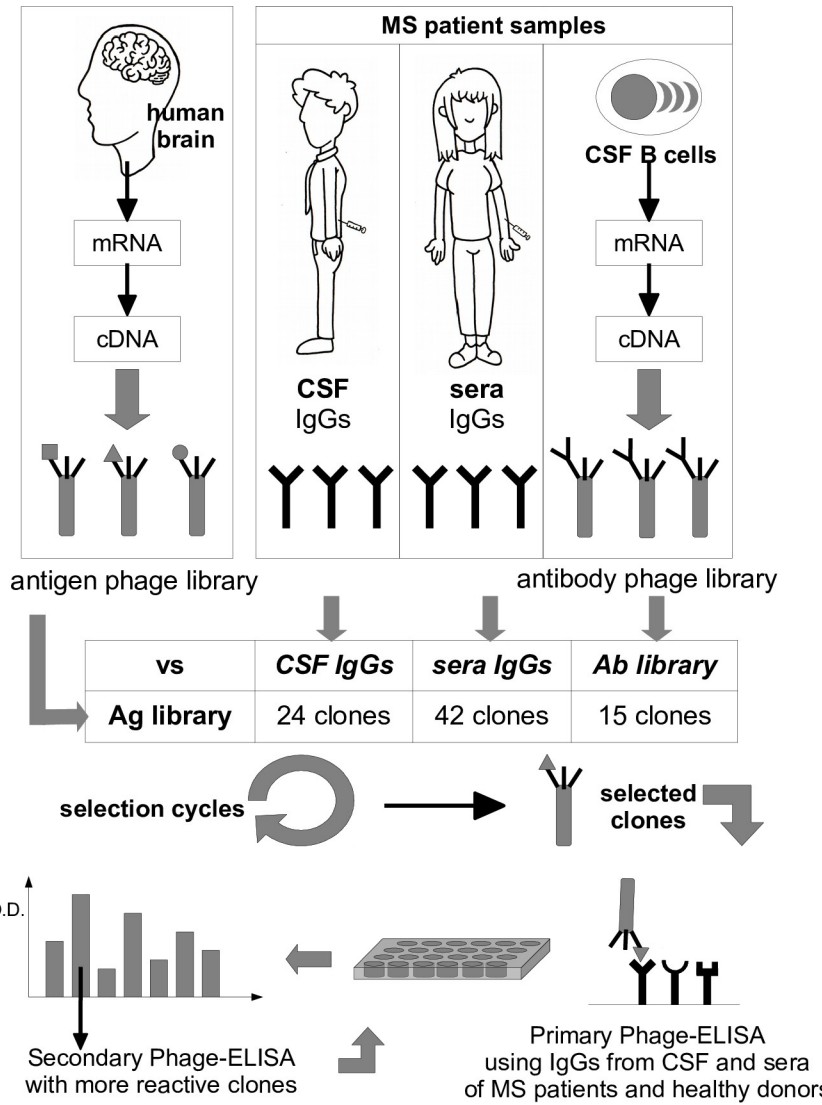

**Fig 1. Overview of the biopanning.** Schematic overview of the phage display selection approach for the identification of candidate autoantigens in multiple sclerosis.

selected as two distinct overlapping clones and one of them also had the highest frequency (4/24).

## Selections of the HB library with IgG from sera of MS patients

We then performed the selection of the HB library using purified IgG from a pool of 90 RR MS sera. After three rounds of selection an enrichment index of 5.4-fold was observed and 190 clones from the last panning were tested by phage-ELISA. The 42 clones (22%) positive in phage-ELISA could be divided into 9 distinct antigenic groups (excluding background clones) as shown in Table 2 (column RR) and S5 and S6 Tables for blastN and blastP analysis. In this selection, the background clones were 30 out of 42 (71%) and almost all the remaining antigens showed the same frequency (1/42); only the DDX24 antigen was present twice on 42. Importantly, some antigens were shared with the selection performed with the pool of CSFs. In particular, the two most frequent peptides selected with the CSFs pool, the antigenic

**Table 2. Summary of putative antigens identified by the selections of the HB phage display library.**

| Antigen | Antigen Symbol | Selection | | | Real ORF |
|---|---|---|---|---|---|
| | | CSF | RR | scFv | |
| Amyloid beta precursor protein | APP | - | 1/42 | - | + |
| ATP synthase membrane subunit e | ATP5ME | - | 1/42 | - | + |
| Brain expressed X-linked 2 | BEX2 | - | 1/42 | - | + |
| DEAD-box helicase 24 | DDX24 | 4/24 | 2/42 | 6/15 | + |
| Erythrocyte membrane protein band 4.1 | EPB41 | 2/24 | - | - | + |
| Flotillin 2 | FLOT2 | 3/24 | 1/42 | 2/15 | - |
| Heterogeneous nuclear ribonucleoprotein U like 2 | HNRNPUL2 | - | 1/42 | - | + |
| Hsp70 interacting protein | ST13 | 1/24 | - | - | + |
| Microtubule associated protein 1B | MAP1B | - | 1/42 | - | + |
| Phosphatidylinositol-4-phosphate 5-kinase type 1 alpha | PIP5K1A | - | - | 2/15 | - |
| Pleckstrin homology domain containing B2 | PLEKHB2 | 1/24 | - | - | + |
| Transcription elongation factor A like 4 | TCEAL4 | - | - | 1/15 | + |
| Transcription elongation regulator 1 | TCERG1 | - | - | 1/15 | + |
| Uridine-cytidine kinase 1 like 1 | UCKL1 | - | 1/42 | - | - |
| Unidentifed antigens / out-of-frame | - | 4/24 | 3/42 | 2/15 | - |
| Background clones | - | 9/24 | 30/42 | 1/15 | |

List of the 14 putative autoantigens identified by the three selections of the HB phage display library. The type of sample (CSF, RR and scFv) used in the selection is indicated (see text for details). The indicated frequency represents the number of clones, among the positive in phage-ELISA, that map to the same antigen. The last column indicates if the clone corresponds to a portion of the open-reading-frame sequence coding for a real protein.

fragments belonging to "DDX24" protein and the Flotilin-2 like peptides were recognized also by the RR sera pool. All together, these data show that, although the use of the serum results in a lower enrichment and a higher number of background clones, the selections with CSF and sera were able to capture the same antigens.

## Selection of the HB library with scFv library

Despite the potentiality of the phage display technology for the identification of autoantigens, in MS the background that could derive from the presence of non-pathogenetic antibodies in the serum and the low titer of Ig present in the CSF of the patients are limiting factors. To overcome these two limitations, we implemented a novel system which involves the selection of the antigen phage-display library with a library of recombinant antibodies. In particular, a single chain (scFv) phage display library of $2x10^4$ independent clones was generated starting from a pool of B cells purified from the liquor of two MS patients. The library diversity was evaluated as usage of unique VL gene per clones and measured as described in Methods. An estimated value of 73% was calculated. It should be noted that, since the B cell present in the CSF of MS patients are a very oligoclonal population, the fact that 27% of the clones would share the same VL fragment is expected and in accordance with previously described data [27]. The scFv library was used to select the HB phage library as described in Methods. Like for the pool of sera and CSFs, three rounds of selection of HB library were performed using the scFv library and 160-fold enrichment of interactive clones was calculated. The specificity of 94 randomly picked clones from the last round of selection was tested by phage-ELISA and the positive clones were characterized by PCR and enzymatic fingerprinting. Out of 15 positive clones (16%) in phage-ELISA we recognised 6 distinct antigens marked by a unique enzymatic fingerprinting pattern, apart from background clones. The results from sequence analysis and

Blast analysis are summarized in Table 2 (column scFv) and S7 and S8 Tables. Noteworthy, the most selected antigen was again DDX24, that was represented as three distinct overlapping peptides with an overall frequency of 6/15 clones. Over 50% (8/15) of the clones were real-ORF clones and, in contrast with the other previous selections, the background level was much lower with only one clone (6%) out of frame. Interestingly, in addition to DDX24 selected with both sera and CSFs, also the clone encoding for the Flotilin-2 like peptide was present among the clones selected with the scFv library.

## Validation of the selected antigens

An initial validation of the selected antigens was performed by phage-ELISA analysing the response of single MS CSF specimens, including the ones used for the selection. As in previous similar study [28] and to investigate the potential use of the antigens as "diagnostic biomarker" for distinguishing MS patients from patients with other neurodegenerative diseases (OND) [29], we decided to compare these MS CSF samples with the response of the CSF of OND patients. Due to the high background observed in the selection with the pool of sera, only the antigens selected with the CSFs and the scFv library were considered for the follow up analysis. Furthermore, we concentrated our work only on the real-ORF clones with the exception of the one translating for the Flotilin-2 like peptide, due to his presence in all three the selections. A first validation was performed using a small cohort of CSFs samples (cohort 1, Table 1) by phage-ELISA. When we tested 12 MS and 8 OND CSFs, the clones carrying the peptides belonging to DDX24 and TCERG1 (Transcription Elongation Regulator 1) were the only ones able to discriminate between MS and OND patients (Fig 2). In particular, MS patients showed a significant higher response towards the two antigens compared to the control group. To confirm these observations, we tested the DDX24 and TCERG1 clones against a second independent larger cohort of CSF samples (cohort 2, Table 1). Both DDX24 and TCERG1 were able to distinguish MS and OND patients (Fig 3); in particular, the difference between MS and OND CSF response in phage-ELISA was more statistically significant using DDX24 than TCERG1 peptide carrying clones.

## Assessment of the diagnostic value of DDX24 and TCERG1

We next investigated if autoreactive antibodies against DDX24 and TCERG1 could be detected in the sera of MS patients and whether their presence could be use as diagnostic test to discriminate between MS and OND. To eliminate the confounding effects that could be derived from the presence of the phage, the two antigens were produced and purified as described in Methods and used to test the reactivity of 18 MS and 19 OND sera (Cohort 3, Table 1) in ELISA. As for the CSFs, also the sera of MS patients showed a significantly higher reactivity against DDX24 and TCERG1 compared to the control group (Fig 4A and 4C).

Receiver operating characteristic (ROC) curve analysis for the DDX24 ELISA showed an area under the curve (AUC) of 0.7646 (95% confidence interval 0.6067–0.9225) and a p Value of 0.005 (Fig 4B). The O.D. cut off of 0.21 yielded the highest Youden's index value for discriminating patients with and without MS at sensitivity of 72.22% (95% confidence interval 46.5–90.3) and specificity of 78.95% (54.4–93.9). At this value, the test exhibited a false positive rate of 12% and a positive predicted value (PPV) of 76.5% (65.5–89.0) with a prevalence weighted likelihood positive ratio (LR+) of 3.43 (4.4–8.6) for the diagnosis of MS. For TCERG1 instead an AUC of 0.9474 (0.8716–1.023) with a p value <0.0001 was calculated (Fig 4D). In this case for an O.D. cut-off of 0.164 the test showed a sensitivity of 100% (81.5–100) and a specificity of 89.47% (66.9–98.7) with false positive rate of 10.5% and PPV of 90% (70.8–97.1) with a LR+ of 9.5 (2.6–35.2).

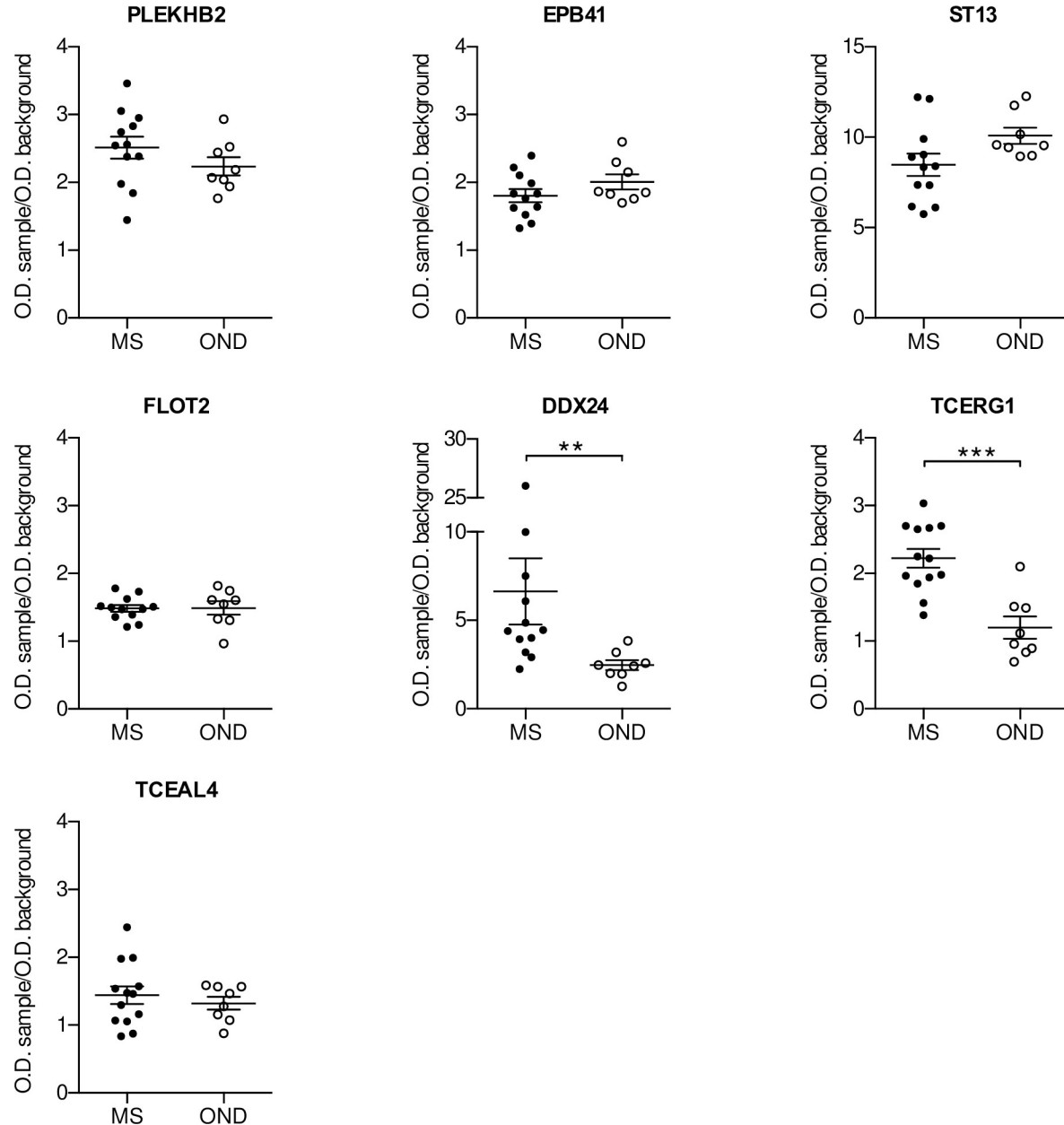

**Fig 2. ELISA validation of putative biomarkers.** Clones isolated by the selection of the HB library were validate by Phage-ELISA using CSFs of MS (n = 12) and OND (n = 8) patients (cohort 1, Table 1). Individual biological replicates are shown. Error bars indicate mean +/- SEM. Mann-Whitney test has been used to assess statistical significance; ** p < 0.01, *** p < 0.001.

We then assessed the possibility to improve the specificity and sensitivity using both antigens in a combined test for the diagnosis of MS (Fig 5). In this case, considering positive only the double positive samples the test showed a sensitivity of 72,2% and a specificity of 100%; for this condition the PPV is 100% and the LR+ tends to infinite. We tested also the reactivity of synthetic peptides obtained from the sequence of DDX24 and TCERG1 against a bigger cohort of sera samples including sera from MS patients utilized in the selections (see S1 Fig for the data). Combining these results as before, the sensitivity was of 43.33% and the specificity of 97.37%; for these values the PPV is 92.86% and LR+ is 16.47. In both analysis using combined

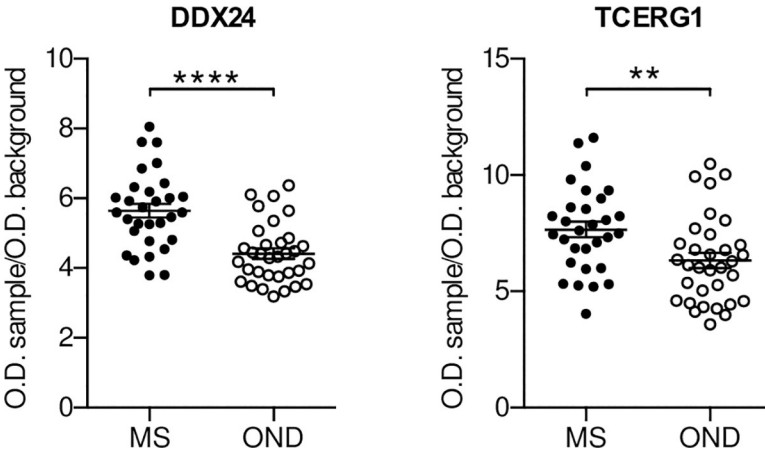

**Fig 3. ELISA validation of TCERG1 and DDX24.** TCERG1 and DDX24 were validate by a secondary Phage-ELISA using a bigger independent cohort (cohort 2, Table 1) of CSFs of MS (n = 30) and OND (n = 33). Individual biological replicates are shown. Error bars indicate mean +/- SEM. Unpaired t-test has been used to assess statistical significance; ** p < 0.01, **** p < 0.0001.

test of two antigens, the sensitivity was low and this could be dependent on the heterogeneity of immune response in MS patients; however the specificity was higher respect to use a single antigen. It should be noted that TCERG1 gave better result to discriminate MS and OND samples as recombinant protein whereas DDX24 as synthetic peptide.

## Discussion

In the past 40 years, the research of the antigens eliciting the B cell response in MS has been the aim of many studies as their identification would be extremely helpful for both a better understanding of MS pathogenesis and its diagnosis. Phage display technology has been one of the methods implemented for the identification of molecular targets of the humoral response in MS [28–30]. Selections of random peptides phage display libraries, using either purified CSF IgG [28] or recombinant antibodies (rAbs) [30] generated from clonally expanded plasma cells present in the CSF of MS patients, were the first approaches used. Despite the data generated with these methods are not easy to interpret as secondary bioinformatic analysis are needed to search putative antigens with high similarity to the peptides selected, these studies led to the identification of several target peptides. However, after further serological analysis no difference in the reactivity towards these antigens could be demonstrated between MS and OND patients [29,31,32]. Somers and colleagues used a more direct system by generating a cDNA phage display library to select possible MS antigens [33]. In their study, the authors select eight reactive clones using IgG purified from a pool of CSFs. Out of the eight clones selected, however, only one corresponded to a known protein expressed in the correct reading frame, while the others were either out of frame or translation of untranslated regions of expressed genes [33].

We have previously reported a new system for the generation of real-ORF enriched cDNA phage display libraries [24]. In the present study, to avoid the biases mentioned above, we have implemented our method to generate a human brain (HB) cDNA phage display library for the selection of MS autoantigens. We selected the HB library using IgG purified from serum or CSF of MS patients and, for the first time, a single-chain variable fragments (scFv) library. As shown in S4, S6 and S8 Tables, after selection percentage of the in-frame clones was

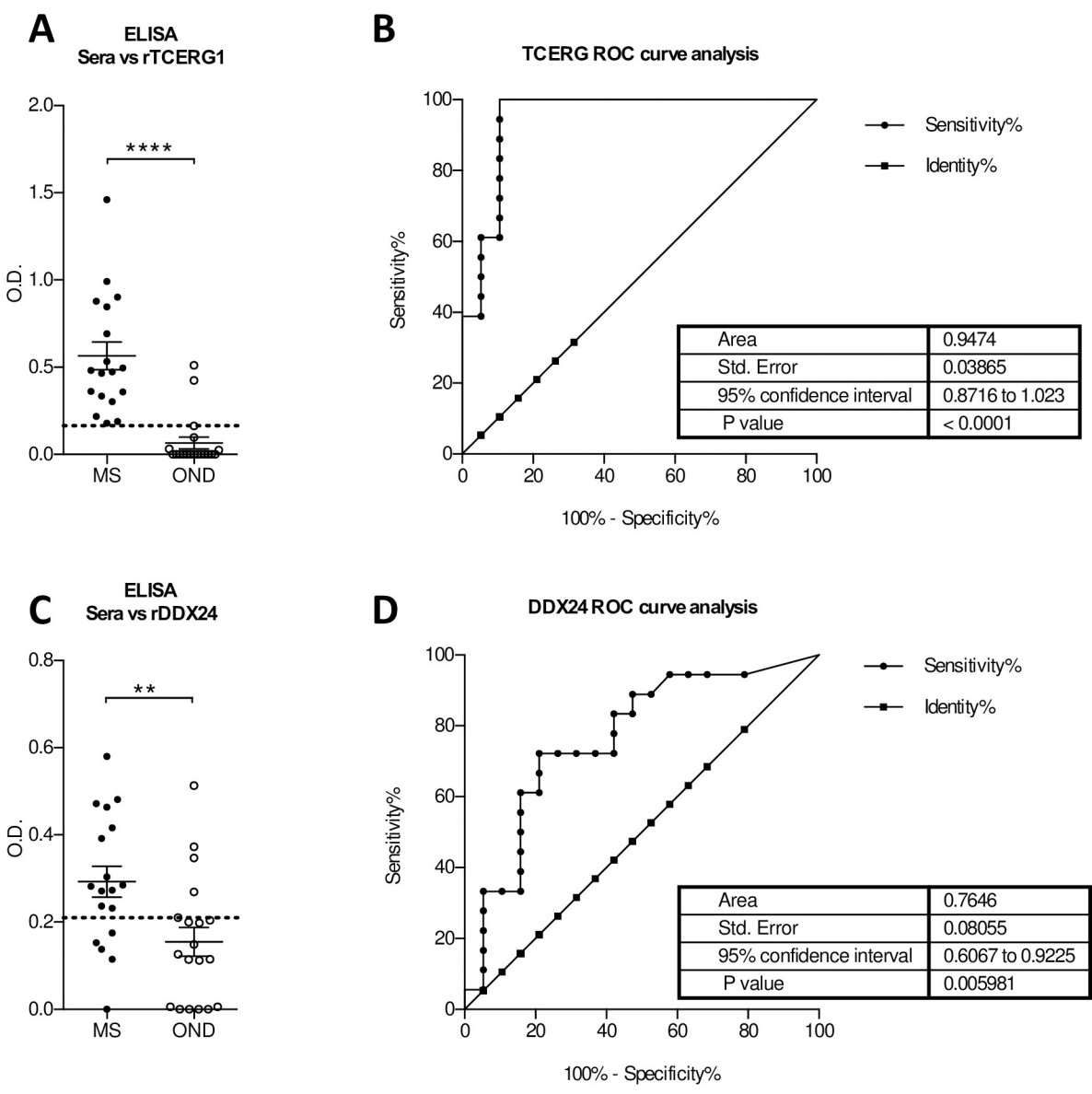

**Fig 4. Evaluation of the diagnostic value of DDX24 and TCERG1 in the prediction of MS.** Serum response against recombinant TCERG1 (A) and DDX24 (C) in MS (n = 18) and OND (n = 19) patients (cohort 3, Table 1) measured by ELISA. ROC analysis of TCERG1 (B) and DDX24 (D) to distinguish MS from OND patients. Dotted lines represent the cut-off with the highest Youden's index. Unpaired t-test has been used in **A** and **C**; ** p < 0.01, *** p < 0.001, **** p< 0.0001.

considerably high varying between 83–93%, as well as the one for the real-ORF clones between 53–71%.

A hallmark of MS is the presence of antibody produced intrathecally by clonally expanded B cells in the CSF of the patients [2]. Hence, a higher specificity of the IgG purified from the CSF is expected, compared to the one from the serum. In agreement to this prediction, the selection with the sera IgGs led to a higher number of putative targets with low frequency compared to the selection performed with the IgG from the CSF where a more restricted number of targets with higher frequency was isolated. The low concentration of IgG in the CSF of MS patients or the low abundance of some specific antibodies could be a limiting factor in the

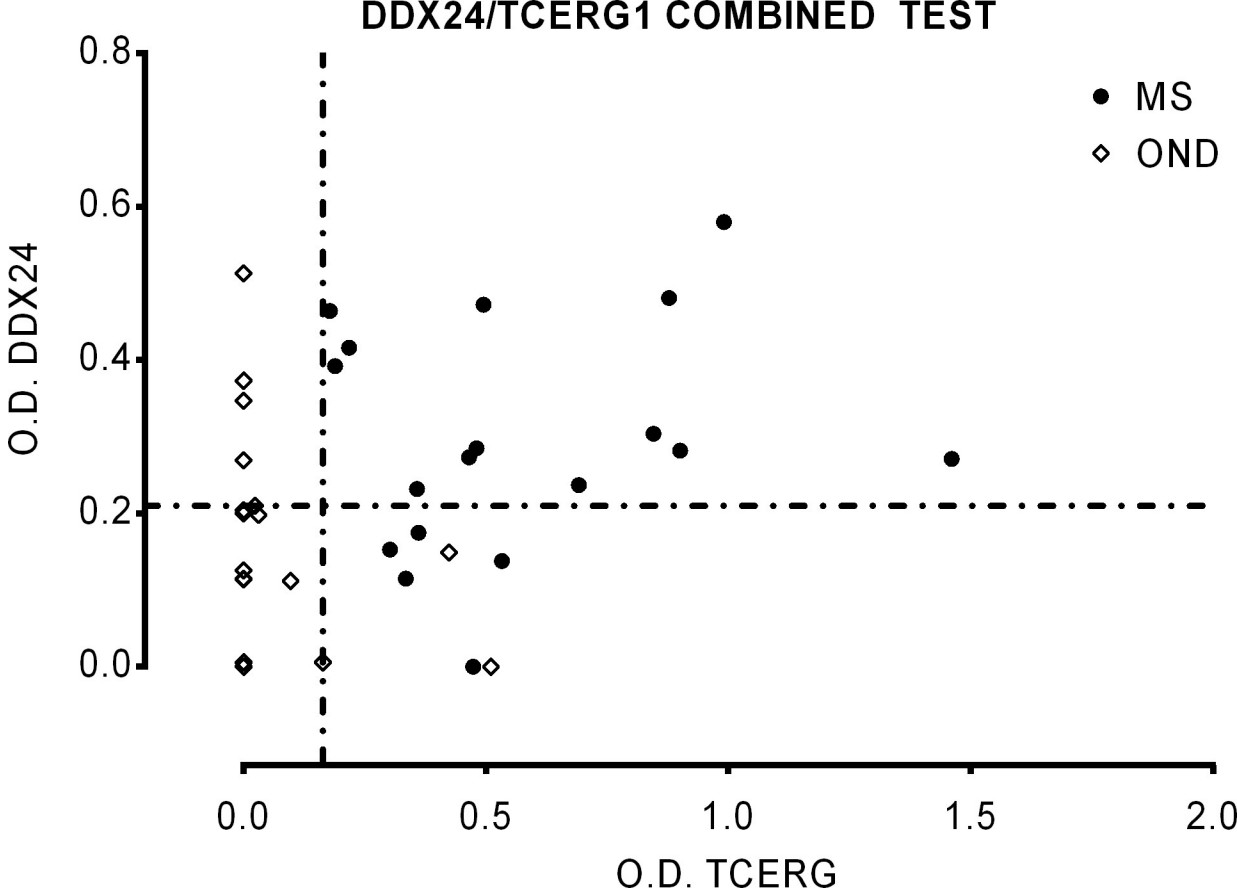

**Fig 5. Evaluation of the diagnostic value of DDX24/TCERG1 combined test in the prediction of MS.** Serum response against recombinant TCERG1 and DDX24 in MS (n = 18) and OND (n = 19) patients (cohort 3, Table 1) measured by ELISA (Fig 4). Individual biological replicates are shown. Dotted lines represent the cut-off with the highest Youden's index calculated for each antigens.

process of bio panning. A possible solution to this problem could be the use of recombinant antibodies generate by cloning the antibody repertoire of the B cells purified from the CSF of MS patients. In their work, Yu and colleagues, have demonstrated the feasibility of using recombinant antibodies to screen phage display random peptide libraries and the cross-specificity between recombinant antibodies and related naïve CSF IgG [34]. Eventually, we used a phage display scFv library, generated by cloning the antibody repertoire of the B cells purified from the CSF of two RR-MS patients, to select the HB cDNA phage display library. Comparison of the three different protocols of selection suggested the benefits of using a library of recombinant antibodies (scFv) for the isolation of putative autoantigens. Selection with the scFv library, in fact, showed the highest percentage (14/15 corresponding to a percentage of 93%) of "in-frame" clones identified (S8 Table) and a good specificity isolating a restricted number of molecular targets with relative high frequency.

Over the last two decades several putative candidates have been identified and proposed as biomarkers that could be useful in the diagnosis and prognosis of MS. Nevertheless, further work is necessary for their validation and application to the clinic and up to date the diagnosis of MS remain based on clinical observation, MRI and presence of oligoclonal bands in the CSF of patients. Therefore the identification of new biomarkers, that could be used to discriminate

between MS and OND, still remain a critical step both for the clinic and to elucidate the mechanism behind the disease pathogenesis [22,23].

Performing a serological analysis of the isolated antigens we identified two new putative autoantigens able to discriminate between MS and OND: DEAD-Box Helicase 24 (DDX24) and Transcription Elongation Regulator 1 (TCERG1). After ROC analysis both antigens show really high sensitivity, specificity and LR+ value (DDX24 72.22%, 78.95%, 3.43; TCERG1 100%, 89.47%, 9.5) and although further independent validations are needed, the data indicates a promising use of the two antigens as biomarker to discriminate MS from OND. Further studies are necessary to better define the correct form of antigen to be used in a supposed diagnostic assay (e.g. peptide or recombinant protein) and to characterize the clinical form of MS that can be discriminated.

DDX24 was isolated with all three protocols and in particular, four different clones representing overlapping peptides of the protein were identified. Of note, the frequency of the DDX24 isolated clones was higher in the selection with the CSF IgG and scFv library and in particular the latter, was the one to isolate three of the four different clones selected. In contrast, TCERG1 was only isolated using the scFv library. Interestingly, despite being more selected, DDX24 showed a less diagnostic value than TCERG1 (Fig 4). These observations together with the enrichment index indicate the better performance of the scFv library in isolating MS-specific autoantigens.

It is now believed that neurodegeneration is the principal cause of permanent disability in MS patients and several evidences have shown as neurodegenerative processes occur throughout the disease course and not only in its progressive phase [2,13–16]. Advance in the study of neurodegenerative disease has revealed the important role of RNA processing and RNA binding proteins (RBPs) in the pathogenic mechanism of these disorders; disruption in RNA expression, splicing and regulation, resulting from abnormal activity of splicing regulatory elements or RBPs, accompanies or drives the neurodegenerative process [35–37].

Interestingly, both antigens that we have identified in this work are protein involved in the regulation of RNA processing and have been associated with neurodegenerative disorders. In particular, TCERG1 has a role in regulating alternative splicing of multiple RNA and its depletion reduces apoptotic processes [38] and leads to alterations in neurite outgrowth [39]. This correlates with the protective effect of TCERG1 overexpression against huntingtin toxicity promoting neuritic aggregates formation [40]. Further TCERG1 increases the expression of TDP-43 which is involved in Amyotrophic Lateral Sclerosis pathogenesis [41]. DDX24 has been implicated in inflammatory diseases and cancer because its role in negative regulation of innate immune response [42] and p53 activity [43]. More recently, whole-transcriptome analysis in patients with intracerebral hemorrhage correlated to neuroinflammation revealed a connection between DDX24 and Sphingosine 1-phosphate receptor 1 that is a target of the treatment of relapsing MS with Fingolimod [44].

One of the possible mechanisms underlying neurodegeneration in MS, and a link to the autoimmune nature of the disease, is the presence in the CSF of patients of autoantibodies targeting neuronal and axonal antigens, in particular against RBPs [12,16]. Autoantibodies against a non-myelin antigen, the heterogeneous nuclear ribonucleoprotein A1 (hnRNP A1), a member of the RBP family, have been detected in the CSF of MS patients [16,45]. Further characterizations have demonstrated that anti-hnRNP A1 autoantibodies can penetrate neuronal cells and target the localization motif of hnRNP. As a result, these Abs can alter hnRNP A1 cellular distribution and induce apoptosis, low levels of ATP, and generation of stress granules; all sign of neurodegeneration [20,45,46]. We therefore can speculate a mechanism by which autoantibodies against TCERG1 and DDX24 might exert a pathogenetic role in MS inducing or sustaining neurodegeneration by impairing the functions of their target proteins either

inducing misplacement or blocking their activity. In this sense these putative antigens could be a marker of a particular disease activity state.

## Supporting information

**S1 Table. Sequence analysis of randomly chosen clones from HB library.** BlastN analysis of 15 randomly picked clones from the ORF-selected phage display cDNA library from human brain. The portion of each identified nucleotide sequence and the percentage of homology are indicated. The sequences of clones that were also coding for the corresponding peptide ("in frame" clones, IF) are indicated; the "out-of-frame" clones (OF) were non coding sequences.
(PDF)

**S2 Table. Antigens selected with the anti-human IgG.** List of antigens identified by the selection of phage display cDNA library from human brain with anti-human IgG.
(PDF)

**S3 Table. Nucleotide sequence alignments of clones selected with CSF pool.** BlastN analysis of 24 positive clones identified by the selection of phage display cDNA library from human brain with pooled and purified IgG from CSF of MS patients. For each clone are indicated: the code of the clone ("Clone"); the clone frequency ("Freq.") corresponding to the number of clones, over the total clones sequenced, that map to the same antigen; the nucleotide sequence identified by blastN analysis ("Identity-blastN"); the NCBI accession number ("GeneBank No.") of the identified sequence; the percentage of homology ("% identity") with the identified sequence; the first and last nucleotide of the identified sequence.
(PDF)

**S4 Table. Protein sequence alignments of clones selected with CSF pool.** BlastP analysis of 24 positive clones identified by the selection of phage display cDNA library from human brain with pooled and purified IgG from CSF of MS patients. For each clone are indicated: the code of the clone ("Clone"); the clone frequency ("Freq.") corresponding to the number of clones, over the total clones sequenced, that map to the same antigen; the aminoacid sequence identified by blastP analysis using the translation of the clone nucleotide sequence ("Identity-blastN"); the NCBI accession number ("ProtBank No.") of the identified sequence; the percentage of homology ("% identity") with the identified sequence; the first and last nucleotide of the identified sequence. The last column reports as the clone can be classified: "ORF" if the clone was in frame and an aminoacid sequence was identified; "mimotope" if the nucleotide and aminoacid sequences identified belong to different gene; "unidentified" if an aminoacid sequence was not identified; "out-of-frame" if the clone was apparently not coding; "background" if the clone was also identified in the selection against only human IgG.
(PDF)

**S5 Table. Nucleotide sequence alignments of clones selected with sera pool.** BlastN analysis of 42 positive clones identified by the selection of phage display cDNA library from human brain with pooled and purified IgG from sera of MS patients. For the meaning of column heading see the legend of S3 Table.
(PDF)

**S6 Table. Protein sequence alignments of clones selected with sera pool.** BlastP analysis of 42 positive clones identified by the selection of phage display cDNA library from human brain with pooled and purified IgG from sera of MS patients. For the meaning of column heading see the legend of S4 Table.
(PDF)

**S7 Table. Nucleotide sequence alignments of clones selected with the scFV library.** BlastN analysis of 15 positive clones identified by the selection of phage display cDNA library from human brain with the scFv phage display library from CSF of two RR-MS patients. For the meaning of column heading see the legend of S3 Table.
(PDF)

**S8 Table. Protein sequence alignments of clones selected with the scFv library.** BlastP analysis of 15 positive clones identified by the selection of phage display cDNA library from human brain with the scFv phage display library from CSF of two RR-MS patients. For the meaning of column heading see the legend of S4 Table.
(PDF)

**S1 Fig. Evaluation of the diagnostic value of pDDX24 and pTCERG1 in the prediction of MS.** The diagnostic value of DDX24 and TCERG1 was further investigated testing the reactivity of some sera samples (30 MS from RR-MS samples utilized in the selections and 38 OND with a mean age of 62 and a ratio of female/male of 14/24) against synthetic peptides pTCERG1 (A) and pDDX24 (C) by an ELISA assay. The synthetic peptides named pDDX24 (aa SQSTAA KVPKKAKTWIPEVHD) and pTCERG1 (aa AAKHAKDSRFKAIEKMKDRE) are included in the aminoacidic portion of antigens recognized in the selections. Unpaired t-test has been used in A and C (**** $p < 0.0001$). A significantly higher reactivity of MS patients against pDDX24 and pTCERG1 compared to the control group was observed (S1A and S1C Fig).
The data of the Receiver operating characteristic (ROC) curve analysis for the pDDX24 and pTCERG1 ELISA are showed near the graph (S1B and S1D Fig). For pDDX4 at O.D. cut off of 0.0765 the sensitivity for discriminating patients with and without MS is of 53.33% (95% confidence interval 34.33–71.66) and specificity of 89.74% (95% confidence interval 75.78–97.13) with a prevalence weighted likelihood positive ratio (LR+) of 5.2 for the diagnosis of MS. For pTCERG1 at O.D. cut-off of 0.055 the test showed a sensitivity of 73.33% (95% confidence interval 54.11–87.72) and a specificity of 81.58% (95% confidence interval 65.67–92.26) with a LR+ of 3.98.
(PDF)

**S2 Fig. Evaluation of the diagnostic value of pDDX24/pTCERG1 combined test in the prediction of MS.** Serum response against synthetic peptides pTCERG1 and pDDX24 in MS (n = 30) and OND (n = 38) patients measured by ELISA (S1 Fig). Individual biological replicates are shown. Dotted lines represent the cut-off with the highest Youden's index calculated for each antigen.
Considering positive only the double positive samples the test showed a sensitivity of 43.33% and a specificity of 97.37%; the PPV and FPR were respectively 92.86% and 2.63% with a LR+ of 16.47.
(PDF)

## Acknowledgments

We are grateful for the suggestions in conceiving the manuscript and support in clinical data to Dr. Cosimo Maggiore and Prof. Marino Zorzon. The article is in memory of Prof. Gilberto Pizzolato.

## Author Contributions

**Conceptualization:** Andrea Cortini, Sara Bembich, Paolo Edomi.

**Data curation:** Andrea Cortini, Sara Bembich, Lorena Marson, Eleonora Cocco.

**Formal analysis:** Andrea Cortini, Sara Bembich.

**Funding acquisition:** Paolo Edomi.

**Investigation:** Andrea Cortini, Sara Bembich, Lorena Marson, Eleonora Cocco.

**Methodology:** Andrea Cortini, Sara Bembich, Paolo Edomi.

**Project administration:** Paolo Edomi.

**Resources:** Eleonora Cocco, Paolo Edomi.

**Supervision:** Paolo Edomi.

**Validation:** Andrea Cortini, Sara Bembich, Lorena Marson.

**Visualization:** Andrea Cortini, Sara Bembich, Lorena Marson, Paolo Edomi.

**Writing – original draft:** Andrea Cortini, Sara Bembich.

**Writing – review & editing:** Andrea Cortini, Eleonora Cocco, Paolo Edomi.

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
