## [Decision Letter · Decision Letter 0]

8 Aug 2019

PONE-D-19-17789

Identification of novel non-myelin biomarkers in multiple sclerosis using an improved phage-display approach

PLOS ONE

Dear Dr EDOMI,

Thank you for submitting your manuscript to PLOS ONE. After careful consideration, we feel that it has merit but does not fully meet PLOS ONE’s publication criteria as it currently stands and that some minor revisions will be required. Therefore, we invite you to submit a revised version of the manuscript that addresses the points raised during the review process.

We would appreciate receiving your revised manuscript by Sep 22 2019 11:59PM. To enhance the reproducibility of your results, we recommend that if applicable you deposit your laboratory protocols in protocols.io, where a protocol can be assigned its own identifier (DOI) such that it can be cited independently in the future. For instructions see: http://journals.plos.org/plosone/s/submission-guidelines#loc-laboratory-protocols

We look forward to receiving your revised manuscript.

Kind regards,

Emanuele Buratti, Ph.D.

Academic Editor

PLOS ONE

Journal Requirements:

1. Thank you for including your funding statement; "The research was partially funded by the Italian multiple sclerosis foundation (Fondazione italiana sclerosi multipla – FISM, www.aism.it) grant n. 2002R26 and 2004R6 to PE. The funder had no role in study design, data collection and analysis, decision to publish, or preparation of the manuscript."

Reviewers' comments:

Reviewer's Responses to Questions

**Comments to the Author**

1. Is the manuscript technically sound, and do the data support the conclusions?

Reviewer #1: Yes

Reviewer #2: Yes

2. Has the statistical analysis been performed appropriately and rigorously? 

Reviewer #1: Yes

Reviewer #2: Yes

3. Have the authors made all data underlying the findings in their manuscript fully available?

Reviewer #1: Yes

Reviewer #2: Yes

4. Is the manuscript presented in an intelligible fashion and written in standard English?

Reviewer #1: Yes

Reviewer #2: Yes

5. Review Comments to the Author

Reviewer #1: In their work Cortini et al. described three strategy to discover new candidate autoantigens of multiple sclerosis (MS). All the strategies use a cDNA library obtained from human brain where the potential new biomarkers are selected using antibody present in bodily fluids of patients (CSF and serum) or by using a recombinant antibody library generated from two MS patients. At the end of their efforts, the authors indicate DDX24 and TCERG1 as a novel autoantigens of interest.

Here are my comments:

Line 118, phrase starting with “Relatively to CSF…” double check the English.

Table 1. I think the term “range” should be removed for the “mean age” column, after all only the mean age is shown in the table.

Line 143. Is it correct that only 1 picogram of poly(A)+ RNA and 2.5 pg of random primers were used for the generation of the library? Similarly, in line 159 the authors describe the use of only 40 pL of cells for electrocompetent cells transformation.

Line 352. How many clones were screened to identify the three “background” antigens? I think it is important to state the number to give more significance to the fact that only three clones were identified. In order to reduce binding background, why the authors did not considered to use anti-IgG antibody to pre-clear the phage population before the selection?

Line 415. What do the authors mean with a scFv antibody library diversity of 73%? How did they estimate the diversity? If 27% of the clones were present more than once in a very small library of 2x104 clones I questioned the quality of the library itself.

Supplementary Table S7. In the legend and in the text, it is mentioned that 15 positive clones were identified, while in the table the clone frequencies are reported to be fractions of 17 clones (i.e. 1/17).

Figure 2. Figure Legend. In the figure for DDX24 there are **** indicating the statistical significance, while for TCERG1 there are **, but in the figure legend the p value is indicated for ** and *** stars.

Line 497. The authors state that “a bigger cohort is needed to calculate this value with a better precision”. Since the described experiment involved the use of recombinant purified proteins, instead of protein fragments displayed on the surface of the phage, the authors should repeat the experiment adding all the other sera utilized in the previous part of the study, where the proteins were displayed on phage, to increase the cohort size and better validate their findings.

Line 550. The authors list the benefits of using recombinant scFv for the isolation of potential autoantigens. I found their conclusions not too convincing. It is known that, when phage selections are performed, the enrichment ratio does not necessarily reflect the quality of a selection. Even in the present work, although the selection performed by using scFv gave a 160-fold enrichment, only 15 out of 94 clones were positive during the ELISA screening (~16%) while 48% of the clones were positive when purified antibodies from CSF were used, with only a 10 fold enrichment. Moreover, the identification of the background clones was obtained selecting the cDNA library against an anti-human-IgG antibody, so the presence of no background when phage scFv directly bound on plastic are used for the selection is not surprise. A selection on an unrelated phage-scFv should have been used to assess the background of this selection strategy.

Reviewer #2: The research article by Cortini et al. described the implementation of a phage display approach to identify new autoantigens in multiple sclerosis (MS).

The authors first generated a cDNA library using RNA from human brain and screened against an IgG pool of cerebrospinal fluid (CFSs) collected from relapsing remitting (RR-MS) and CIS patients. The antigen library was also screened against an antibody scFv library obtained from RNA of B cells purified from CFS of two RR-MS. Within this approach, they selected 7 potential autoantigens that were validated in a first cohort of RR-MS and patients with other neurological disease (OND). Among them, two autoantigens, namely DDX24 and CTERG1, resulted significantly increased in CFS form RR-MS. Then, validation in a third cohort of RR-MS patients found that autoantibodies against DDX24 and CTERG1 were increased also in the periphery (serum) of RR-MS. Thus, the authors suggest DD24X and CTERG1 as new potential biomarkers for MS.

General comment: This manuscript is of impact and very interesting for its translational potential. It would be, even more impacting, to test these biomarkers in PP-MS patients where neurodegeneration is an important part of the process, ectopic follicles are found within the brain, and the identification of specific biomarkers is still an unmet need.

The manuscript is well written. Overall, my recommendation is to accept it with minor revisions.

Minor comments:

1- To generate the scFv library the authors selected two drug-naïve RR-MS patients. Where the RR-MS patients of the validation cohort without any treatment as well? Please specify in the text

3- A figure showing a schematic overview of the generation of library/screening would be helpful for readers

4- Abbreviation and acronyms should be specified the first time they appear in the text. I found the full name of the two new autoantigens DDX24 (DEAD-Box Helicase 24) and TCERG1 (Transcription Elongation Regulator 1) at the end of the discussion. Please correct all along the text for all abbreviations.

5-lines 593-608: the authors should rephrase this part, because in this study they are not demonstrating any pathogenetic mechanism about the effect of these autoantibodies. They might change the word “propose” with “can speculate”

6- In Figure legend 1 the p value is missing

7-line 781: number 2 is repeated twice

6. PLOS authors have the option to publish the peer review history of their article (what does this mean?). If published, this will include your full peer review and any attached files.

Reviewer #1: Yes: Fortunato Ferrara

Reviewer #2: No

---

## [Author Response · Author response to Decision Letter 0]

30 Sep 2019

Response to reviewers

Reviewer #1

1) Line 118, phrase starting with “Relatively to CSF…” double check the English.

The sentence has been corrected.

2) Table 1. I think the term “range” should be removed for the “mean age” column, after all only the mean age is shown in the table.

The term “range” has been removed for the “mean age” column, as correctly suggested.

3) Line 143. Is it correct that only 1 picogram of poly(A)+ RNA and 2.5 pg of random primers were used for the generation of the library? Similarly, in line 159 the authors describe the use of only 40 pL of cells for electrocompetent cells transformation.

The use of “p” for pg and pL at lines 143 and 159 was a typographical error and it has been corrected in µg and µL.

4) Line 352. How many clones were screened to identify the three “background” antigens? I think it is important to state the number to give more significance to the fact that only three clones were identified. In order to reduce binding background, why the authors did not considered to use anti-IgG antibody to pre-clear the phage population before the selection?

To identify the three “background” antigens 40 clones have been screened. As suggested by reviewer, this data has been included in the text.

About the suggestion to preclear the phage library, it is necessary to specify that the construction of the library from human brain was a very complex procedure, in particular for the initial quantity of RNA available and we preferred to use the library directly for the selection. A pre-clearing step has been evaluated and will be included in the future improvement of the protocol. Nevertheless the results from the background analysis were convincing.

The related sentence has been corrected

5) Line 415. What do the authors mean with a scFv antibody library diversity of 73%? How did they estimate the diversity? If 27% of the clones were present more than once in a very small library of 2x104 clones I questioned the quality of the library itself.

We thank the reviewer for the comment and we agree that a better explanation should be given. We express the scFv antibody library diversity as percentage of unique VL fragment in our library, which we evaluated by PCR followed by enzymatic fingerprinting (line 178 Material and Methods) of random clones. 

It should be noted that since the B cell present in the CSF of MS patients are a very oligoclonal population, the fact that 27% of the clones would share the same VL fragment is expected and in accordance with previously described data (Owens et al. J Immunol 2003).

We have revised the related sentence.

6) Supplementary Table S7. In the legend and in the text, it is mentioned that 15 positive clones were identified, while in the table the clone frequencies are reported to be fractions of 17 clones (i.e. 1/17).

The number 17 in Supplementary Table S7 was a typographical error and it has been corrected in 15.

7) Figure 2. Figure Legend. In the figure for DDX24 there are **** indicating the statistical significance, while for TCERG1 there are **, but in the figure legend the p value is indicated for ** and *** stars.

The correct value is p < 0.0001, correspondent to four asterisks, and it has been corrected in the legend.

8) Line 497. The authors state that “a bigger cohort is needed to calculate this value with a better precision”. Since the described experiment involved the use of recombinant purified proteins, instead of protein fragments displayed on the surface of the phage, the authors should repeat the experiment adding all the other sera utilized in the previous part of the study, where the proteins were displayed on phage, to increase the cohort size and better validate their findings.

We agree with the suggestion of the reviewer but a clarification about the guide lines on the use of multiple sclerosis patients samples could be useful. Using MS sera in diagnostic assays, as ELISA tests, it is preferred to employ fresh sample or unfrozen one time at most. For this reason, we could not use all the sera utilized in the selection also in the validation tests; further in some cases the available quantity was very low. 

In any case, as suggested, we have performed another ELISA analysis with a bigger cohort of samples utilized in the selections; since, in the meantime, we have tested some synthetic peptides correspondent to the recognized portion of the identified antigens, we preferred to use this type of antigen in the assay, in particular for DDX24 antigen for which it is necessary to denature the protein.

Consequently we have included a Figure that describes the ELISA assay as Supplementary material and we have revised the sentence about the combined test using the recombinant proteins.

9) Line 550. The authors list the benefits of using recombinant scFv for the isolation of potential autoantigens. I found their conclusions not too convincing. It is known that, when phage selections are performed, the enrichment ratio does not necessarily reflect the quality of a selection. Even in the present work, although the selection performed by using scFv gave a 160-fold enrichment, only 15 out of 94 clones were positive during the ELISA screening (~16%) while 48% of the clones were positive when purified antibodies from CSF were used, with only a 10 fold enrichment. Moreover, the identification of the background clones was obtained selecting the cDNA library against an anti-human-IgG antibody, so the presence of no background when phage scFv directly bound on plastic are used for the selection is not surprise. A selection on an unrelated phage-scFv should have been used to assess the background of this selection strategy.

We agree with the reviewer that the fold enrichment cannot be correlated with a better selection; however, we think that our observations support our statement on the advantage to perform selection with the scFv library and a better clarification on the classification of clones (S4, S6 and S8 tables) might help to explain our conclusions. All the clones classified as ORF, mimotope or “unidentified antigens” correspond to “in-frame” clones being their translated sequence without stop codons. Of consequence, although it is true that the CSF selection lead to 48% of the clones positive in Phage-ELISA, of these only 62,5% (15/24) were clones with an “in-frame” insert while the others were background clones (S4 table). In contrast, although the selection with scFv lead to have only 16% of positive clones in phage-ELISA, of these 93% (14/15) were “in-frame” clones (S8 table). Moreover, one of the antigens (TCERG1), able to discriminate MS from OND patients, has been selected only with scFv library indicating that the use of this system could overcome limitation in the use of Ig directly from the CSF where the low concentration of IgG in the CSF of MS patients or the low abundance of some specific antibody could be a limiting factor in the selection of specific autoantigens.

To better explain these observations and accepting the suggestion of the reviewer, we revised the sentence where we described the classification of the clones and we modified the sentence about the advantages of selection with scFv.

Reviewer #2

1) To generate the scFv library the authors selected two drug-naïve RR-MS patients. Where the RR-MS patients of the validation cohort without any treatment as well? Please specify in the text

Also the RR-MS patients of the validation cohort were without any treatment. This has been specified as suggested.

2) A figure showing a schematic overview of the generation of library/screening would be helpful for readers

A figure showing a schematic overview of the generation of library and screening has been included as suggested by the reviewer and consequently all the figures has been renumbered. A sentence in the section Results has been included.

3) Abbreviation and acronyms should be specified the first time they appear in the text. I found the full name of the two new autoantigens DDX24 (DEAD-Box Helicase 24) and TCERG1 (Transcription Elongation Regulator 1) at the end of the discussion. Please correct all along the text for all abbreviations.

The abbreviations were checked throughout the entire text. The full name of the two new autoantigens DDX24 and TCERG were correctly specified the first time they appear, as correctly requested.

4) lines 593-608: the authors should rephrase this part, because in this study they are not demonstrating any pathogenetic mechanism about the effect of these autoantibodies. They might change the word “propose” with “can speculate”

We agree with the reviewer and the sentence has been changed as suggested.

5) In Figure legend 1 the p value is missing

The p value in Figure legend 1 has been indicated.

6) line 781: number 2 is repeated twice

The typographical error has been corrected.

---

## [Decision Letter · Decision Letter 1]

9 Oct 2019

PONE-D-19-17789R1

Identification of novel non-myelin biomarkers in multiple sclerosis using an improved phage-display approach

PLOS ONE

Dear Dr EDOMI,

Thank you for submitting your manuscript to PLOS ONE. After careful consideration, we feel that it has merit but does not fully meet PLOS ONE’s publication criteria as it currently stands. Therefore, we invite you to submit a revised version of the manuscript that addresses the two very minor final points raised during the review process (changing a sentence and adding a bibliographic reference).

We would appreciate receiving your revised manuscript by Nov 23 2019 11:59PM. To enhance the reproducibility of your results, we recommend that if applicable you deposit your laboratory protocols in protocols.io, where a protocol can be assigned its own identifier (DOI) such that it can be cited independently in the future. For instructions see: http://journals.plos.org/plosone/s/submission-guidelines#loc-laboratory-protocols

We look forward to receiving your revised manuscript.

Kind regards,

Emanuele Buratti, Ph.D.

Academic Editor

PLOS ONE

Reviewers' comments:

Reviewer's Responses to Questions

**Comments to the Author**

1. If the authors have adequately addressed your comments raised in a previous round of review and you feel that this manuscript is now acceptable for publication, you may indicate that here to bypass the “Comments to the Author” section, enter your conflict of interest statement in the “Confidential to Editor” section, and submit your "Accept" recommendation.

Reviewer #1: (No Response)

Reviewer #2: All comments have been addressed

2. Is the manuscript technically sound, and do the data support the conclusions?

Reviewer #1: Yes

Reviewer #2: Yes

3. Has the statistical analysis been performed appropriately and rigorously? 

Reviewer #1: Yes

Reviewer #2: Yes

4. Have the authors made all data underlying the findings in their manuscript fully available?

Reviewer #1: Yes

Reviewer #2: Yes

5. Is the manuscript presented in an intelligible fashion and written in standard English?

Reviewer #1: Yes

Reviewer #2: No

6. Review Comments to the Author

Reviewer #1: I appreciated how the authors addressed all my comments.

I still have a couple of minor comments:

- Line 120 in the revisited manuscript:

"For the selection against 121 CSF samples, were pooled samples of 7 untreated RR MS patients (EDSS 0-3.5) and 4

122 clinically isolated syndrome (CIS) patients (EDSS 0-2)"

I am pretty confindent that the correct English version of such sentence should be:

"Samples from 7 untreated RR MS patients (EDSS 0-3.5) and 4 122 clinically isolated syndrome (CIS) patients (EDSS 0-2) were pooled for the selection against 121 CSF samples".

-Line 431. I think it is worth to mention the citation of Owen et al. 2003 to briefly explain the oligoclonal nature of the antibodies obtained from CSF of MS patients.

Reviewer #2: The authors have satisfactorily responded to all my questions and my recommendation is : Accept With No Changes

7. PLOS authors have the option to publish the peer review history of their article (what does this mean?). If published, this will include your full peer review and any attached files.

Reviewer #1: Yes: Fortunato Ferrara

Reviewer #2: No

---

## [Author Response · Author response to Decision Letter 1]

19 Nov 2019

Reviewer #1

- Line 120 in the revisited manuscript:

"For the selection against CSF samples, were pooled samples of 7 untreated RR MS patients (EDSS 0-3.5) and 4 clinically isolated syndrome (CIS) patients (EDSS 0-2)"

I am pretty confindent that the correct English version of such sentence should be:

"Samples from 7 untreated RR MS patients (EDSS 0-3.5) and 4 clinically isolated syndrome (CIS) patients (EDSS 0-2) were pooled for the selection against CSF samples".

We agree with the reviewer and the sentence has been corrected. 

-Line 431. I think it is worth to mention the citation of Owen et al. 2003 to briefly explain the oligoclonal nature of the antibodies obtained from CSF of MS patients.

The bibliographic reference has been cited and the following sentence has been added.

Old Sentence

An estimated value of 73% was calculated. The scFv library was used to select the HB phage library as described in Methods. 

New sentence

An estimated value of 73% was calculated. It should be noted that, since the B cell present in the CSF of MS patients are a very oligoclonal population, the fact that 27% of the clones would share the same VL fragment is expected and in accordance with previously described data [27]. The scFv library was used to select the HB phage library as described in Methods.

The references of the citations has been revised accordingly.

---

## [Editor Report · Decision Letter 2]

21 Nov 2019

Identification of novel non-myelin biomarkers in multiple sclerosis using an improved phage-display approach

PONE-D-19-17789R2

Dear Dr. EDOMI,

We are pleased to inform you that your manuscript has been judged scientifically suitable for publication and will be formally accepted for publication once it complies with all outstanding technical requirements.

With kind regards,

Emanuele Buratti, Ph.D.

Academic Editor

PLOS ONE
---

## [Editor Report · Acceptance letter]

26 Nov 2019

PONE-D-19-17789R2 

Identification of novel non-myelin biomarkers in multiple sclerosis using an improved phage-display approach 

Dear Dr. Edomi:

I am pleased to inform you that your manuscript has been deemed suitable for publication in PLOS ONE. Congratulations! Your manuscript is now with our production department. 

With kind regards,

on behalf of

Dr. Emanuele Buratti 

Academic Editor

PLOS ONE